# "If it wasn't forced upon me, I would have given it a second thought": Understanding COVID-19 vaccine hesitancy in an outlier county in the Bay Area, California

**Carinne Brody** *, **Julian Szieff**, **Bilal Abu-Alrub**

Public Health Program, College of Education and Health Sciences, Touro University California, Vallejo, California, United States of America

* cbrody@touro.edu

**Data Availability Statement:** Study participants were not informed through our consent process that the data from this study would be held in a repository or be made available to the public

## Abstract

While the San Francisco Bay Area counties rank very low in vaccine hesitancy and high in vaccination rates when compared to national numbers, Solano County has the most residents who are hesitant (6.3%) and the least who are fully vaccinated (51.6%) among Bay Area counties, according to the CDC. No studies to date have been able to provide the in-depth, localized information that would inform county-level public health interventions. This study aims to explore reasons and consequences for remaining unvaccinated in Solano County. Using a qualitative in-depth interview approach, we conducted 32 interviews with unvaccinated adults from Solano County. We used a grounded theory approach to our analysis. Using the socio-ecological model as a framework, we found that reasons for not getting vaccinated were primarily structural (mistrust of public information) and individual (bodily autonomy, personal choice) while consequences were primarily interpersonal (discrimination and stigma from friends, family, and employers). An overarching theme was that the vaccine rollout and messaging felt like an encroachment on personal choice and the feeling of being forced created more mistrust. Participants reported feeling like their decisions made them minorities among their colleagues, friends and family and that they were not persuaded by groupthink or by their relationships. Future public health responses to epidemics and pandemics might consider if a vaccine mandate is the best approach for reaching all county residents.

## Introduction

During the 2020–2022 COVID-19 pandemic, San Francisco Bay Area counties ranked low in COVID-19 vaccine hesitancy and high in vaccination rates when compared to national numbers. However, by August 2022 Solano County had the smallest proportion of residents who are fully vaccinated (70%) compared to surrounding counties such as San Francisco (86%), Contra Costa (85%) and Marin (88%) [1]. Research is still unfolding about reasons for COVID-19 vaccine hesitancy.

through publication. Our Institutional Review Board has thus concluded that this would be an unethical use of data and has denied our request to publish the transcripts of this study. However if specific researchers would like to view transcripts they can be made available on a case-by-case basis. We have supplied our codebook as supplement material. Point of contact at our Institutional Review Board is Dr. Sahai Burrowes; 707-638-5837; sburrowe2@touro.edu.

**Funding:** This study was funded by the Touro University California's Intramural Research Award Program (IRAP) (IRAP 2021 Brody).

**Competing interests:** The authors have declared that no competing interests exist.

Recent research from early in the pandemic gathering some information about vaccine hesitancy in North America. Survey-based studies have found that vaccine hesitant groups are made up of people who are more female, have less vaccine knowledge, have more belief in vaccine conspiracies, have a perception of COVID-19 as low severity, are more likely to be African American and Hispanic, more likely to have children at home, rural dwellers, lower household income, people in the northeastern U.S. and those who identified as Republicans [2, 3], and among healthcare workers they were more likely to be younger and less educated [4]. A Canadian content analysis study conducted after vaccines were made available analyzed COVID-related tweets by Canadians. The themes identified in the 605 tweets included in the study were: concerns over safety, suspicion about political or economic forces driving the COVID-19 pandemic or vaccine development, a lack of knowledge about the vaccine, anti-vaccine or confusing messages from authority figures, and a lack of legal liability from vaccine companies [5]. These studies provide important evidence for public health efforts to understand vaccination rates.

Several qualitative and mixed methods studies from around the world offer some evidence of perceptions and reasons for hesitancy. Studies of university students in Canada [6] Israel Defense Forces Soldiers [7] and racially diverse UK healthcare workers [8] have shown that different groups are hesitant for different reasons ranging from assessment of person risk for young people to mistrust based on historical injustices and abuses of power. One study conducted before the vaccine was made available used semi-structured phone interviews gather perceptions of US-based Latino families about COVID vaccines to inform public health messaging. Almost half of respondents expressed hesitancy citing mistrust, fear and the political nature of the vaccine development [9]. We were unable to identify any qualitative studies exploring reasons for vaccine hesitancy among the general population of unvaccinated people in the US since the vaccine has been made available to the public. We found one interview-based study on vaccine hesitancy among women leaving jails in Kansas. They found that health literacy was low while mistrust, misinformation, and conspiracy theories were common [10].

There is no clear evidence that tells us what works to understand reasons for and consequences of COVID-19 vaccine hesitancy. Nationally, while workplace mandates and incentives have boosted rates temporarily, they have not proven to increase rates to the needed levels to reach herd immunity. Some research suggests that even transparent messages with detailed information about COVID-19 vaccines did not significantly reduce vaccine hesitancy [11]. Previous research from public health, behavioral economics and marketing suggests that tailoring interventions to regional or local trends is a practical strategy that can be successfully implemented. They recommend segmenting the population based on their specific "identity barriers" (personal or political identity) and feelings of "uniqueness" (I have different needs than others that are being neglected)–themes that can be identified through qualitative research [9].

Solano County is a racially diverse county north of San Francisco with a population size of 448,747 people. The county is 34.4% White, 29.1% Hispanic, 14.4% African-American, 17.4% Asian, and 7.5% Mixed Race. During the COVID-19 pandemic, the local media called Solano County an outlier in the Bay area in terms of vaccination rates citing personal and political identity barriers. In early 2020, Solano County health officials first resisted shelter-at-home orders publicly even though it was the first county in the nation to report a case of coronavirus that couldn't be traced to overseas travel or contact with an infected person. Solano county was the last bay area county to move out of the red tier (high level of transmission) in June 2021. In Fall 2021, while neighboring Alameda County was making it mandatory to show proof of vaccine to enter restaurants, Solano supervisors voted down a proposal requiring that county employees be vaccinated, saying it should be a personal choice and only unvaccinated

people should be required to wear masks indoors [12]. At that time, Solano County was the only local county not requiring everyone to wear a mask when indoors in public. A Solano County's health officer had said publicly that indoor mask mandates didn't have any benefit for the counties that did enforce restrictions [13]. In contrast, the public health director had been strongly in support of masking and vaccinations. In the wake of the first infant death from COVID in the region in December 2021 he said, "I urge Solano residents to get vaccinated, or boosted, for COVID-19. Getting the COVID vaccine is the most effective approach we can take to protect our families and communities, most especially the children who are too young to get vaccinated" [14]. These events demonstrate the lack of a clear message to the public during the two years of the pandemic which county official attribute to unclear state guidance [15, 16].

This study aims to explore reasons and consequences for remaining unvaccinated in Solano County.

## Materials and methods

This qualitative research study used purposive sampling to recruit Solano County residents that self-report being unvaccinated for in-depth interviews by phone or video call. Inclusion criteria included being a current Solano County resident, deciding not to get the COVID-19 vaccine and being able to participant in a phone or video call for up to one hour. Three research assistants (RAs) were hired as recruiters and interviewers and were trained by the PI. After training, the RAs conducted pilot interviews as an assessment of their readiness for data collection and were then given feedback on their technique and areas for improvement. The interview guide was amended after pilot testing to include several specific prompts for questions 1–3 (for example, "Was your choice to remain unvaccinated influenced by anything else in your life?*),* an additional question about impacts on job and career since the decision not to be vaccinated and the addition of "Is there anything more you would like to add?" at the end of the interview.

Research assistants were given a flyer (paper and electronic) with recruitment information that they hand out to potential participants with an email address for contacting the research staff. Participants were offered a $100 visa card for their participation. Recruitment flyers were handed out by RAs at public places throughout the county including all public libraries, civic centers, community centers, farmers markets and transportation hubs. In addition, it was posted on local private and public Facebook groups including "Vallejo Needs," "Solano County Community Awareness," "What's happening in Solano County," and "What's going on in Solano County" as well as Nextdoor.com. Once a potential participant contacted the research staff by email, they were asked to submit an online consent form and schedule a one-hour interview by either phone, zoom with video or zoom without video. Before the interview began, they were again informed of the purpose of the study, risks and benefits and then asked for their verbal consent to continue. Interviews were conducted during April and May 2022.

Based on a review of the literature and discussion with the research team, an interview guide was developed that covers a range of COVID-19 vaccine topics including beliefs about the pandemic, acceptance and practice of containment measures, beliefs and attitudes about COVID-19 vaccines, behavior drivers and sources of information. The interview guide was piloted with a convenience sample of 2 unvaccinated county residents and revised based on pilot participant feedback.

A modified grounded theory approach was used to guide analysis of the transcripts– grounded theory is the generation of theory which is 'grounded' in data that has been systematically collected (Glaser and Strauss 1967). It was modified in that we did not use line-by-line coding but instead formed concepts from general interpretations of the data. We used post-

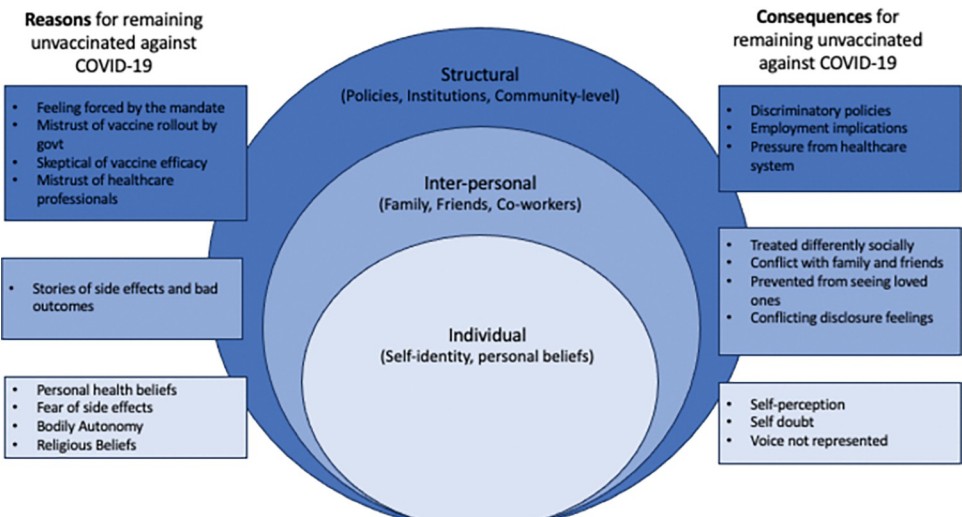

**Fig 1. The structural, interpersonal, and individual level reasons and consequences of remaining unvaccinated.**

interview memos and constant comparison methods to guide our data collection and inter-views. We constructed codes inductively and did not initially use a pre-existing theory to guide our study. We saw that there emerged many different levels of reasons and consequences for vaccine hesitancy that were similar to a classic ecological model. Due to these levels, we then referred to the socio-ecological model to help structure our final codebook.

Interview recordings were transcribed by Zoom's transcription feature and refined by hand. No identifying information was available during analysis. Qualitative data was analyzed first with in vivo coding and then secondary thematic coding using Dedoose (Dedoose qualita-tive coding software, Version 8.0.35, 2018). Two researchers independently labeled sections of transcripts and created initial codes. The researchers then came together to discuss mis-matches in codes. Then the team refined codes and sub-codes until they felt that no new codes were occurring in the data, also known as saturation. This process of coding iteratively resulted in a final codebook. The codebook was used to develop a diagram that summarizes the concep-tual relationships between the codes, or a conceptual model, that aids in understanding the experience of the participants in a novel way (Fig 1).

We then summarized our findings and shared them with all participants via email. We included a form for participants to provide feedback on what they read. We received 8 responses and have integrated the quotes from their feedback forms into the results below.

## Ethical considerations

This study underwent ethical review and approval by the Touro University California Institu-tional Review Board (TUC IRB Application #PH-0222). All key researchers involved in this study have completed a CITI certification course on the protection of human research partici-pants. The records (recordings, reports, transcripts, and memos) of this study will be kept con-fidential. The research team did not include the name or personal identifiers of any participants involved in the research in our reports or within transcripts. The phone interview recordings, transcripts and memos are kept on in a password-protected folder only accessible by our research team. The principal investigators and the research assistant will be the only ones to have access to these records. We will keep these documents in this secure folder for 3 years after the completion of the research per federal guidelines.

## Results and discussion

We completed 32 interviews with an average interview length 31.25 minutes. Through our analysis identified the below themes and codes. In Fig 1, we present a conceptual diagram which displays our themes and codes using a modified socio-ecological model. We categorized our findings into structural, interpersonal and individual level reasons for and consequences of remaining unvaccinated. We describe each theme and code and give example quotes.

### Reasons

The following themes are categorized under reasons that participants gave for remaining unvaccinated for COVID-19.

**Individual reasons.** *Personal health beliefs*. Many participants felt that they didn't need the COVID-19 vaccine to protect their health. There were three main divisions of motivation among this group. Many believed that because they are healthy, their immune systems could fight off COVID without the vaccine. Some had already recovered from COVID and felt they had natural immunity.

> "I'm not diabetic. I'm not obese. My blood pressure is fine, my heart is fine, so I didn't have comorbidities that would lead me to a place where I think I should really get this." *Participant 132*

Other participants mentioned feeling no need for the vaccine because of their regular practice of alternative preventative measures against COVID-19 such as mask-wearing, hand washing, use of hand sanitizer, healthy diet, and being cautious in public. A few participants cited feeling confident in the available treatment options if they were to get COVID.

> "If I were to catch COVID, it would not be good because of my asthma. And that scares me. But I think in the back of my mind because my doctors told me, I qualify for the antibody treatment. If I were to catch COVID, that makes me feel better." *Participant 108*

*Fear of side effects*. Many participants were concerned about the possibility of experiencing side effects from the vaccine.

> "You're going to hear stuff like myocarditis, the inflammation of the heart. That was a big thing a couple of months ago. Now that's a reported side effect, and they need to make it clear that in very rare instances, people are going to get sick and may even die from the symptoms." *Participant 121*

> "You would show all these people that were dying from COVID. But you were not showing these millions of people dying from the vaccine. So, it was just untruthful. I knew someone personally. Right after they got the vaccine, suddenly their heart started having problems and then they died." *Participant 123*

These concerns for side effects also extended to concern for children. Participants tended to have this concern because they felt that the vaccine was made too quickly and current research regarding its efficacy and long-term effects are insufficient.

> "I guess I feel scared to get it. I mean, my son is four years old right now, but he'll be turning five in July. So, it just makes me scared. . .if I get him the vaccine, you know, how will his body take it? How? You know, there's not any studies, I feel like we aren't far out enough

and don't have like the years of studies behind the vaccine that I just feel like scared on the way that it might affect my son." *Participant 114*

A notable subgroup of participants concerned with the side effects of the vaccine were particularly concerned with how the vaccine would react with a current or previous medical condition.

"I belong to support groups for my illness. . .a lot of people are arriving who had never had our illness before and now have it after getting vaccinated. Or people who have been in remission who no longer were in remission after getting their vaccine. That scares me." *Participant 118*

*Bodily autonomy*. Participants talked about the belief that getting the vaccine was an infringement on their bodily autonomy. These individuals chiefly cited not wanting to put something unnatural and potentially harmful into their bodies as well as wanting to have control over what happens to their body.

"I don't want to and I don't need to fix anything that's not broken. It's just something in my body that I don't need." *Participant 129*

"But it's the same thing where my body, my choice, and so it's my body, my choice. I felt that it was inappropriate and completely wrong for anyone to try and dictate what I do with my body." *Participant 122*

*Religious beliefs*. A small group of participants cited their religious beliefs as an important motivating factor for not receiving the COVID-19 vaccine. These participants noted trust in their faith rather than medicine.

"That really, in my heart, I believe God was protecting me, protecting my family, and protecting my children. Not to just do something and jump on board when they have you know, absolutely no information and to more trust in God, trust in my prayers." *Participant 105*

**Interpersonal reasons.** *Stories of side effects / bad outcomes*. Participants talked about the stories of side effects and bad outcomes about the COVID-19 vaccine that they had heard. Participants reported accounts from friends, family members, and coworkers regarding the vaccine's negative effects.

"Because the other thing is my wife's cousin who is post-menopausal. She's 60 years old. She had a period for a month after mRNA injections. So, I mean it's not like I just read about it. I actually know two people that that happened to." *Participant 132*

Many of these participants pointed out that vaccinated individuals still contracted COVID-19 and experienced symptoms. Some noted that unvaccinated individuals largely have the same experience after contracting COVID-19.

"And but what I have experienced is the majority, a lot of people that are vaccinated, including co-workers, . . . got COVID more than people that I know that weren't vaccinated. I had a few about four of my co-workers, they were all vaccinated, double vaccinated, and

they've gotten a positive COVID test and their family members. They were like, you need to get it, you need to get it, and they were the ones that came down with COVID." *Participant 116*

**Structural reasons.** *Feeling forced by the mandate*. One of the most mentioned reasons for not getting vaccinated among participants was negative feelings towards feeling forced to receive the COVID-19 vaccine. The idea of free will regarding medical decisions came up frequently.

"I felt that it was inappropriate and completely wrong for anyone to try and dictate what I do with my body, and as, I think, as a semi-intelligent person who can weigh things out, I was able to make my own decision. Therefore, I did." *Participant 122*

"I don't like to be forced into it or coerced. So, if they were able to make me think like it was my idea "Maybe I should go get the vaccine", then maybe I would have been quicker to get it. But since it was like "You need to get the vaccine. . .it's like teenagers, right? You tell them go left, they go right. If it wasn't forced upon me, I would have given it a second thought." *Participant 101*

Some of the interviewees who mentioned not wanting to be forced stated they would have been more willing to get vaccinated if the choice to do so was their own and they were not pressured from someone else.

"It was just the mandate. It should be a medical decision. And should be between the patient and the doctor. And the government should not be involved in that. So, I'm just stubborn enough." *Participant 121*

*Mistrust of vaccine rollout by government/science community*. Most participants felt mistrust of the COVID-19 vaccine rollout. The three main reasons participants cited for this mistrust were lack of convincing information, concern about the financial motivation and corruption of the government and pharmaceutical companies and aggressive persuasion tactics that made the vaccine seem like a scam.

Within the group of participants who felt there was a lack of convincing information regarding the vaccine's efficacy, there were four distinct sub-groups. Most participants within this subset noted that current COVID-19 research available regarding the vaccine's efficacy and side effects was insufficient at the time when participants were interviewed.

"Yeah, I am saying that the public service information, that's telling people hey, we created the vaccine suddenly and all you got to do is get to vaccinating and that will save you. Yes, I agree that it appears that people who have a vaccine are less likely to die from the disease and fewer of them need hospitalization, but we don't know the long-term impact of gene therapy. That has been minimally tested on human populations." *Participant 119*

"I do think you captured what most people had to say, and I am in there a bit, but your data summary sounds like the research subjects structurally mistrust medical authority. For me, that is incorrect because I generally do trust medical authority. Only in the case of the COVID gene therapy passed off as a vaccine do I balk. It was done too fast; it was not sufficiently tested; it was marketed globally with no right to sue the producers for damage; and

the companies have been allowed to overcharge and threaten and abscond with wealth of the countries that may try to find workarounds." *Feedback Form 8*

Most participants also felt there was inconsistent public messaging about the vaccine. These participants frequently reported that the changing news about boosters and decreased preventative effects over time undermined their belief in the vaccine's safety, efficacy, and necessity.

"It's just the inconsistency. Like wear a mask, don't wear a mask. Oh, it just seems like it's not just public health, it's related to how many businesses are lying to our governments about the financial impact that they're having. And no inside dining. Inside dining. To go everything. It's either a public health crisis or not. You can't just bend back and forth for financial reasons or political reasons." *Participant 102*

Some participants felt that they personally did not have enough information about the vaccine to make an informed decision.

"I would say it's scary, not having all the information one wants. It's scary to go with what they're offering. They're still not set in stone into what is needed and what's not in regards to the vaccine." *Participant 115*

A significant portion of interviewees discussed concern about government and pharmaceutical company corruption and questioned who was benefiting from mass vaccination campaigns. These participants felt the vaccine was pushed because the government and pharmaceutical companies were benefiting financially from its systemic distribution.

"Big Pharma is really kind of scary and untrustworthy. And then when a lot of money is involved and power and corruption. All of that is kind of just swept the rug, and I think we haven't for a long time had faith in the government or Big Pharma. They screw up all the time with things, so it's hard to trust." *Participant 129*

A smaller portion of interviewees discussed scam-like persuasion tactics that contributed to their vaccination hesitancy. Public announcements about rewards such as cash or free groceries for getting vaccinated were particularly perceived as deceptive.

"Another place offered like 50 bucks to take the vaccine. . .for you to get the shot. . . . just as a way to make people take the shot. Just a way to sucker people into taking the shot." *Participant 125*

*Skeptical of vaccine effectiveness*. Some participants pointed out that vaccinated individuals still contracted COVID-19 and experienced symptoms. These participants noted that unvaccinated individuals largely have the same experience after contracting COVID-19.

"And but what I have experienced is the majority, a lot of people that are vaccinated, including co-workers . . . got COVID more than people that I know that weren't vaccinated. I had a few about four of my co-workers, they were all vaccinated, double vaccinated, and they've gotten a positive COVID test and their family members. They were like, you need to get it, you need to get it, and they were the ones that came down with COVID." *Participant 116*

"The efficacy of the COVID-19 vaccines and other measures have been proven to be less than they were purported to be. I'm [unvaccinated], obese, borderline diabetic and have

asthma, but I wasn't hospitalized when I had covid. It was deceptive to say that they would prevent infection before the data proved that claim. People lost their jobs due to fear, miscommunication, and misinformation. How will that be addressed now that the truth about vaccines and boosters are public?" *Feedback Form 4*

*Mistrust of healthcare profession.* Some interviewees felt pressure from the medical system and felt that they could not fully trust their healthcare providers. Historical discrimination against people of color and those in poverty was mentioned by some participants.

"And then there's also the mistrust, in a lot of communities of color they just don't trust the healthcare system. Just a lot of historical implications. And just the level of healthcare in Vallejo. I don't know about Fairfield, but Vallejo, our healthcare system is pretty lackluster, so we are probably not rushing to go there." *Participant 102*

"I feel like they do advertise to us, to people of color because most of the time we're poor. And we don't have good medical resources and they just want to put the pressure on us because they think all poor people are stupid, or some of us are not educated. That's why I will not take it."

Participant 106

Others mentioned not feeling heard by their doctor as well as being concerned that the healthcare system does not care about them.

"Some of these doctors and these health care workers don't believe in the vaccine but it's their job to push it, it's their job to make the patients feel comfortable getting it." *Participant 108*

**Consequences.** The following themes are categorized as consequences that participants experienced for remaining unvaccinated for COVID-19.

**Individual consequences.** *Self-Perception/not an anti-vaxxer.* Multiple participants expressed they were not against vaccines and that being labeled as an anti-vaxxer felt unfair. Some stated they felt that given the information that was presented to them getting the vaccine was not in their best interest and that it was their right to choose.

"I mean the fact that we call it "unvaxxed" I think is wrong. Because I've gotten every vaccine that has ever been recommended to me, except for this. So the word "unvaxxed" is very unfair because I absolutely believe in vaccines. But I don't believe in unproven medications that we don't know the long-term side effects of. I don't think that's a good idea to be injected into my body. It might be perfectly fine. But I'm not going to do it. I think it's actually very degrading."

Participant 122

"I generally trust vaccines, and have always taken them, but now I worry that even a flu shot might be mixed with genetic nanoparticles or mRNA just to get that material inside my body. That is not an unreasonable fear." *Feedback Form 6*

*Self-doubt.* Some participants expressed doubts about their decision to not get vaccinated. These concerns revolved around wondering if their choice was the correct one and how their

decision would impact those around them in the community and within their own families. A few participants expressed how this wore on their mental stability as well.

"...I hugged her... And then after the fact, you know I'd be a little anxious like, Oh, my goodness, I hope I didn't make her ill or whoever it was. And am I doing the right thing? Am I doing the right thing? Am I being selfish? Am I being unreasonable? All these myriad of thoughts and emotions. I've been up and down with them these past couple of years." *Participant 107*

*Voice not represented*: Some participants felt that they were never given the opportunity or space to talk about why they chose to remain unvaccinated. Some participants reported that the language used to describe the unvaccinated population created a feeling of ostracization.

"It really feels like we're living in a time where the community, or the world even, is divided by vaccinated verse unvaccinated. It's all over social media. It's this thing and I'm just like, man. We need a voice. We need people to speak up, and actually hear us out and hear our beliefs in our position on this. And maybe, like I said, all around, people could have more compassion." *Participant 111*

**Interpersonal consequences.** *Treated different socially*. The most reported consequence of being unvaccinated was being treated differently socially by others. This theme captured a variety of perceived negative social experiences that differed from pre-pandemic interactions.

"When they found out I was unvaccinated, they didn't want to be around me. They thought I was crazy. They would just give me these looks. And I was not allowed to go into businesses. Being unvaccinated is not equal to being sick or being a carrier, I mean I'm not Typhoid Mary. I wore a mask, kept my distance, washed my hands, sprayed everything down, did everything else but get a vaccine. *Participant 132*

"I want to respect everybody's decision and what they want. Like if you don't want me at your house because I don't have my vaccine, I understand and respect that. But I don't understand why you can't talk to me because I don't have the vaccine. Talking to me is not harming you." *Participant 109*

Similarly, other participants reported negative social interactions such as being perceived as stupid or uncaring and being looked down upon for their vaccination status.

"...just the constant barrage of making those of us that decided not to get the vaccine—making us feel bad about our decisions. It was interesting to see how, especially in the media, how we were painted to be people that were either misinformed or just downright stupid by thinking that there was gonna be some type of you know GPS device that was being put into our body."

Participant 122

A few participants reported that their children were subjected to social ramifications for being unvaccinated against COVID-19 as well. One of these participants described a situation where they were dropping their child off at a birthday party when they were met by a woman at the door asking if the child had been vaccinated against COVID-19.

*Conflict with family, friends, and colleagues.* Many participants reported experiencing conflict with those close to them, including family, friends, and colleagues. One participant reported having to perpetually justify his decision to remain unvaccinated.

"I had family members that were always "Don't come around me" and "Are you vaccinated?". It was crazy. Even sometimes you go around certain people and they'll say, "Are you vaccinated" and I'm like "No." And they're like "Go get a mask." It's just with time you get used to it now. At first it was just "Wear your mask, boy". It's an interesting journey. But the strongest pressure comes from family and they're like "Why?" and they have the right to just question you with everything. "Why aren't you doing that?" Why, why, why. It's like "I don't want to. I don't trust it" "Why?" I'm not going into all that. The pressure was just a lot more intense."

Participant 123

Another participant discussed COVID-19 as a catalyst in pushing an already strained relationship with their mother-in-law past the point of civility.

"But with my mother-in-law we had a blowout and didn't talk for probably a year because she told me I was a bad parent, that I didn't want to get vaccinated. She lost her mind on me when she she asked me over the phone if I was gonna vaccinate my boys. And I said, no, absolutely not. And we it was a huge fight. I already have like a strained relationship with her. So I don't know why she chose to talk to me about vaccinating my kids. But it turned into such a huge fight. And we didn't talk for a really, really long time. . ."

Participant 108

*Prevented from seeing family and friends.* Many participants reported being unable to see family members and friends due to their vaccination status. Some of these participants discussed being unable to see grandparents or family members who were chronically ill due to concerns about them contracting COVID-19.

". . . they [family) don't want to touch any more like you have some like walking disease or something, you know, it's just weird. So, we haven't even been to my grandma's house."

Participant 116

Multiple participants expressed how difficult extended seclusion from their families was and the toll it took on them.

"I was feeling kind of alone. I didn't have anyone even though I did have family, because I wasn't seeing them and because it was kind of hard for me to adjust and not just go in my car and go to see my aunt or my cousins or my sisters. So it was really hard for me not be able to physically see them." *Participant 109*

*Conflicting disclosure feelings.* Another unique aspect of the COVID-19 pandemic was the weight disclosure of vaccination status carried. Multiple participants expressed the negative consequences of disclosing their vaccination status to people in the community or in the workplace. Some felt that this information was unnecessary to share and remained quiet or led others to believe that they were vaccinated to avoid stigma.

"I don't disclose it. Because I just feel like it creates an awkward situation. For example, I worked at the schools for a little bit and everybody in the staff room was vaccinated, and they'd constantly have these conversations about how angry they were with people who weren't vaccinated and how irresponsible they were and how you know, it was the unvaccinated people's fault for things not going good and they should just take the damn vaccine because, you know, it's gonna help everyone. And so the majority of the people I was around, were always talking about how they had just gotten the vaccine booster, so I didn't really chime in on those conversations." *Participant 108*

Other participants admitted to directly lying in order to feel more accepted amongst the vaccinated or to gain entrance to public venues.

"Fine. I just lied. "I left my card at home". It's easy. It's a joke, like you have to be vaccinated in order to go in a bar or a restaurant and all that, but you can just lie. They don't care, they really, really do not care. They care? You just go someplace else. It's fine. It's easy. Nobody cares. Because it's a personal medical decision." *Participant 121*

**Structural consequences.** *Employment Implications.* Some participants discussed their frustration with their vaccination status putting their income at risk. Both in the context of applying for jobs and maintaining existing jobs participants described feeling pressure from their employers.

"I did have a recent job interview. I mean the guy that was interviewing me seemed a little bothered that I wasn't vaccinated but you know he kept bringing it up in the interview. Like hey you need to be vaccinated, you should be vaccinated. I mean he gave me the job but I ended up like not working anyway. Because he really wanted me to get the vaccination and that was kind of a problem. . ." *Participant 106*

One participant noted that since his work was funded by grants from the government, workers had to adhere to COVID-19 vaccination mandates for their projects to receive funding.

*Faced discriminatory policies.* The predominant theme regarding structural consequences for participants was facing discrimination such as being denied access by an organization or business to public activities. Many participants reported being unable to enter and participate in public activities such as dining at restaurants, attending sporting events, and being involved in school activities.

"And a lot of places like San Francisco require you to be vaccinated. So it just felt like there was a lot of things that I couldn't go do. And so that was a big push for people to get vaccinated like, "Oh, if you want to have an actual life. . ." Or like my daughter, she's 13. She wanted to go to a concert in Chico, and we had to be vaccinated. So it's like "Oh, well sorry, we can't go do certain things anymore because we are being forced." *Participant 129*

"As far as going out, socializing, and all that kind of stuff, I couldn't do anything because I wasn't vaccinated. I've been rejected at bars, clubs. Been rejected. Made complete plans and was rejected at the door because it didn't have enough information." *Participant 123*

One participant described the difficulty of testing perpetually for COVID-19 to bypass the vaccination requirements.

"...we didn't get to go to "Disney on Ice" and we missed Warriors games and Giants games, like simply because I'm not vaccinated. And we've had instances, you know, a lot of places, if you're not vaccinated, you can get tested, and you have to show a negative result within like a window of time. So multiple times, I would go get tested and feel like, okay. we're gonna get these results. And I didn't even get the results until days after the event. So we missed the event." *Participant 108*

One participant discussed being barred from seeing their wife, who was living in a nursing home, due to their unvaccinated status.

Another participant who is a health care worker provided a unique perspective regarding their experience with discrimination. This participant noted how their contribution to healthcare during the pandemic, where they routinely risked their health, appeared to be overshadowed by their unvaccinated status.

"I felt very unappreciated as a healthcare worker, working through the pandemic long hours, being subjected to work in hazardous environments with little to inadequate levels of PPE protection to then be later down the line characterized as a villain or black sheep, subjected of disciplinary action for refusing to be vaccinated." *Participant 102*

*Pressure from the healthcare system*. Some participants discussed feeling pressure from the healthcare system as unvaccinated individuals. These participants reported being uncomfortable, not listened to, and disrespected as they received healthcare.

"I did go see an optometrist. He kind of gave me that attitude like you're not vaccinated? I strongly urge you to get vaccinated. Kind of looking down at you. You didn't do what you were supposed to do." *Participant 102*

Some participants expressed understanding that doctors need to discuss vaccination but were frustrated because they felt that their decision to remain unvaccinated was not respected.

"He [the doctor] wanted to nail down every reason that I had. You can call it cognitive dissonance. You could call it whatever you want. I can explain to you in a way that you will understand and accept my reasons for doing it. And at some point, I was like, thank you. After about four conversations. And I mean these were intense. Like I thought these were like debates that we were having and I didn't feel he . . .like he didn't respect my decision. He didn't respect my right to make that decision." *Participant 111*

This study found that participants' main reasons for choosing not to get the COVID-19 vaccine were based on their personal beliefs about how to protect their health and their skepticism of the vaccine's effectiveness and safety. Recent studies support these findings. A lack of confidence in the vaccine's effectiveness and concerns about the safety of the vaccine were main decision motivators among unvaccinated individuals [17–20]. Both emphasis on personal choice and distrust of the government and pharmaceutical companies were frequently cited reasons to not get vaccinated. Other studies also found distrust of the development of the vaccine [4, 10, 21–23]. Other studies have found that those who decide to not get vaccinated also highly value autonomy [17]. Our findings suggest that participants' vaccine decisions were not swayed by consequential discrimination, job loss, or interpersonal conflict.

Participants describe the main consequences of being unvaccinated as interpersonal issues. Participants discussed social as well as physical isolation and the accompanying mental health

consequences. Studies from other populations have shown that unvaccinated individuals have higher levels of psychological distress, including depression, anxiety, and insomnia [24, 25]. Participants in our study described the interpersonal conflicts they experienced with family interactions, childcare, healthcare, and at the workplace. While there are no other studies that discuss interpersonal conflict as a result of not getting the COVID vaccine, there are studies that describe parents' experiences of social exclusion as a result of not getting their child their routine childhood vaccines [26].

Participants recounted that public health messaging and restrictive policies were discriminatory and stigmatizing. Other studies have found that messaging around many aspects of the COVID pandemic including social distancing and disease transmission may have been designed to protect the community but may in fact be experienced as discriminatory [27, 28]. We know that past responses to epidemics such as HIV, leprosy and Ebola have used discriminatory messaging which worked counter to public health goals [29–31]. The lesson from these studies is that approaches to COVID-19 vaccine information campaigns should be carefully designed to reduce stigma and discrimination and promote inclusion.

Our qualitative approach captured wide ranging and detailed interviewee responses describing the reasons and consequences associated with the decision to be unvaccinated. This study is one of very few studies that have investigated the decision-making process and personal experiences of individuals who are unvaccinated against COVID-19, particularly those who live in or near highly vaccinated areas and must navigate vaccination requirements.

There are some limitations to consider for this qualitative study. This study utilized self-selection for participant recruitment through advertisement fliers and posts. This method could have excluded those with stronger negative feelings towards the COVID-19 vaccine because they may be less likely to choose to participate in university-sponsored research. In addition, the authors are from public health and medical backgrounds and designed the study to minimize bias which impacts the interpretation of interview transcripts. Individuals from a different background may have interpreted these transcripts differently. The data of this study was collected in the spring of 2022 in Solano County after initial COVID-19 boosters were announced and available, but before the second round of boosters became available. Participants' feelings about the vaccine and boosters may have changed since the data was collected. Finally, the qualitative nature of this study means that our findings are not intended to be generalizable.

## Conclusion

This study found that vaccine hesitant participants felt that the vaccine was unsafe and ineffective and the pressure to get vaccinated felt like a violation of personal choice. Participants expressed that autonomy in their healthcare decisions was deeply important to them and these decisions were not changed by the discrimination and interpersonal conflict they experienced. Future public health responses to pandemics might better serve the public by focusing on building trust, creating intentionally non-stigmatizing campaigns and developing policies that mitigate the mental health implications for those that may be late adopters or decide against mandated public health measures. Preparedness may also include the use of novel methods such as attitudinal inoculation which is a preventative approach to combating misinformation through pre-exposure to misinformation and delivery of information that allow individuals to refute it [32].

## Supporting information

**S1 Checklist.** *PLOS ONE* **clinical studies checklist.**
(DOCX)

**S1 File. Initial codebook.**
(PDF)

# Acknowledgments

Thank you to our additional interviewers and transcribers Taylor Moss, Kathleen Mak and Stephanie Wong.

# Author Contributions

**Conceptualization:** Carinne Brody.

**Data curation:** Julian Szieff.

**Formal analysis:** Carinne Brody, Julian Szieff, Bilal Abu-Alrub.

**Funding acquisition:** Carinne Brody.

**Methodology:** Carinne Brody.

**Project administration:** Carinne Brody.

**Writing – original draft:** Carinne Brody, Julian Szieff, Bilal Abu-Alrub.

**Writing – review & editing:** Carinne Brody.

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
