## [Decision Letter · Decision Letter 0]

28 Jun 2023

PONE-D-23-13758“If it wasn’t forced upon me, I would have given it a second thought”: Understanding COVID-19 Vaccine Hesitancy in a Diverse County in CaliforniaPLOS ONE

Dear Dr. Brody,

Thank you for submitting your manuscript to PLOS ONE. After careful consideration, we feel that it has merit but does not fully meet PLOS ONE’s publication criteria as it currently stands. Therefore, we invite you to submit a revised version of the manuscript that addresses the points raised during the review process.

 Overall, the manuscript is well-written and organized.  The research question is novel, the authors applied flawless qualitative methods and reported findings effectively.   

I recommend the authors to provide a table of demographics for participants and assign a participant ID to include in the quotes provided in the results section. I also recommend the authors to report the average length for the interviews conducted.

We look forward to receiving your revised manuscript.

Kind regards,

Mohammad Nusair, Ph.D

Academic Editor

PLOS ONE

Additional Editor Comments:

I enjoyed reviewing this manuscript. Overall, the manuscript is well-written and organized. The research question is novel, the authors applied flawless qualitative methods and reported findings in an effectively.

I recommend the authors to provide a table of demographics for participants and assign a participant ID to include in the quotes provided in the results section. I also recommend the authors to report the average length for the interviews conducted.

Reviewers' comments:

Reviewer's Responses to Questions

**Comments to the Author**

1. Is the manuscript technically sound, and do the data support the conclusions?

Reviewer #1: Yes

Reviewer #2: Yes

2. Has the statistical analysis been performed appropriately and rigorously? 

Reviewer #1: N/A

Reviewer #2: N/A

3. Have the authors made all data underlying the findings in their manuscript fully available?

Reviewer #1: Yes

Reviewer #2: Yes

4. Is the manuscript presented in an intelligible fashion and written in standard English?

Reviewer #1: Yes

Reviewer #2: Yes

5. Review Comments to the Author

Reviewer #1: This is a very interesting paper.

I was easy to read and well documented. The flow should be commended. It kept the attention of the reader.

A solid and detailed qualitative analysis. The methods were well explained.

It addresses the "consequences", which I found innovative. Usually only "reasons" are addressed.

Some suggestions:

1. In the title the word "diverse" is used to qualify the county. However, in the paper there is no explanation of the sociodemographic and other characteristics of the Solano County. That could help illustrate why it is diverse. Please expand.

2. I suggest that the under "reasons for remaining unvaccinated...", the individual category of "personal beliefs" be recoded as "personal health belief" or any other term that allude to the perception of personal health. The quotes and explanation included in the manuscript refers to this dimension of personal belief.

Good job!

Reviewer #2: Dear Authors,

It was a pleasure to review your work. I congratulate you for addressing such an important topic using the approach you used. Although overall I like you work, please consider the following recommendations:

Title: you used quotation marks on the title, and in qualitative research, that indicates that that statement in quotation marks comes from the study participants. I could not find that specific quote in your work. Also, the title needs revision since it does not capture the full scope of your work. Please consider the use of ‘diverse’ as part of your title.

Purpose of the study: the aim of the study is clearly stated in the abstract; however, it should be clearly stated in the manuscript itself. The last part of the introduction could be an excellent place to include it.

Introduction: on the introduction of the second paragraph, you referenced one Canadian study that analyzed 605 tweets. You closed that paragraph, referring to all studies as quantitative. Please, clarify if the Canadian study was also quantitative since it does not seem like it.

Methods:

o please include the inclusion criteria

o Was the ecological model used to develop the questions guide?

o What were the revisions of the question guide after the pilot? Those are important to share.

o The paragraph before the last one on the method sections needs revision. The verbs term needs to be revised. See if you can give more clarity on this paragraph.

o You mentioned a diagram that ‘summarizes the conceptual relationships between the codes.’ Consider sharing that diagram since that can add clarity or transparency to this section.

Results:

o It is unclear if you collected demographic information about your participant, but if you did, consider adding a table with that information since it will provide valuable information to your study. Also, is information that can contribute to the analysis.

o It will be good if you add at the beginning of the result section that you used the ecological model to organize your result section.

o In addition to the participant id, it will contribute to your paper and help to better understand the phenomenon if you add any demographic not identifiable information to the quotes you presented.

Discussion and conclusion:

o An aspect for consideration is to expand on the discussion or conclusion about the aim of public health. Public health responses aim to benefit groups of people, entire populations, and more during a pandemic. You are evaluating the impact of public health measures on individuals, which is very important. However, it would also be valuable to add the aim of public health efforts, not avoiding the impact of those measures on the individual.

Best wishes!

6. PLOS authors have the option to publish the peer review history of their article (what does this mean?). If published, this will include your full peer review and any attached files.

Reviewer #1: No

Reviewer #2: No

---

## [Author Response · Author response to Decision Letter 0]

17 Jul 2023

Response to Reviewer (PONE-D-23-13758)

I enjoyed reviewing this manuscript. Overall, the manuscript is well-written and organized. The research question is novel, the authors applied flawless qualitative methods and reported findings in an effectively.

RESPONSE: Thank you.

I recommend the authors to provide a table of demographics for participants and assign a participant ID to include in the quotes provided in the results section. I also recommend the authors to report the average length for the interviews conducted.

RESPONSE: Thank you for this comment. We did not collect demographic information due to concern about trust and confidentiality for this particular group. We do have the average interview length (31.25 mins) which we added to the results section.

Reviewer #1: This is a very interesting paper. I was easy to read and well documented. The flow should be commended. It kept the attention of the reader.

A solid and detailed qualitative analysis. The methods were well explained. It addresses the "consequences", which I found innovative. Usually only "reasons" are addressed.

RESPONSE: Thank you for this feedback.

Some suggestions:

1. In the title the word "diverse" is used to qualify the county. However, in the paper there is no explanation of the sociodemographic and other characteristics of the Solano County. That could help illustrate why it is diverse. Please expand.

RESPONSE: Thank you, we discussed and think that a better more descriptive word in the title would be “Outlier” and “Bay Area”. We have also included sociodemographic statistics of Solano County. 

2. I suggest that the under "reasons for remaining unvaccinated...", the individual category of "personal beliefs" be recoded as "personal health belief" or any other term that allude to the perception of personal health. The quotes and explanation included in the manuscript refers to this dimension of personal belief.

RESPONSE: We have changed this section to “personal health beliefs”.

Good job!

Reviewer #2: Dear Authors,

It was a pleasure to review your work. I congratulate you for addressing such an important topic using the approach you used. Although overall I like you work, please consider the following recommendations:

Title: you used quotation marks on the title, and in qualitative research, that indicates that that statement in quotation marks comes from the study participants. I could not find that specific quote in your work. Also, the title needs revision since it does not capture the full scope of your work. Please consider the use of ‘diverse’ as part of your title.

RESPONSE: Thank you, we discussed and think that a better more descriptive word in the title would be “Outlier” and “Bay Area”. We have also included sociodemographic statistics of Solano County. We have also added the example quote from the title – thank you for this recommendation. It is under “Feeling forced by the mandate” and reads: “I don’t like to be forced into it or coerced. So, if they were able to make me think like it was my idea “Maybe I should go get the vaccine”, then maybe I would have been quicker to get it. But since it was like “You need to get the vaccine, there’s (inaudible)”. It’s like teenagers right? You tell them go left, they go right. If it wasn’t forced upon me, I would have given it a second thought.” Participant 101

Purpose of the study: the aim of the study is clearly stated in the abstract; however, it should be clearly stated in the manuscript itself. The last part of the introduction could be an excellent place to include it.

RESPONSE: We have added the sentence from the abstract to the end of the introduction section.

Introduction: on the introduction of the second paragraph, you referenced one Canadian study that analyzed 605 tweets. You closed that paragraph, referring to all studies as quantitative. Please, clarify if the Canadian study was also quantitative since it does not seem like it.

RESPONSE: Thank you, this was a content analysis and therefore not quantitative. We have omitted that word.

Methods:

o please include the inclusion criteria

RESPONSE: We have added this under materials and methods: “Inclusion criteria included being a current Solano County resident, deciding not to get the COVID-19 vaccine and being able to participant in a phone or video call for up to one hour”

o Was the ecological model used to develop the questions guide?

RESPONSE: No, we did not use this model to develop the guide. We developed an initial list of codes and used the model to better structure our codebook when we saw levels emerging. We have added this language to clarify: We saw that there emerged many different levels of reasons and consequences for vaccine hesitancy that were similar to a classic ecological model. Due to these levels, we then referred to the socio-ecological model to help structure our final codebook.

o What were the revisions of the question guide after the pilot? Those are important to share.

RESPONSE: We have added the things that changed after pilot testing to the material and methods section: “The interview guide was amended after pilot testing to include several specific prompts for questions 1-3 (for example, “Was your choice to remain unvaccinated influenced by anything else in your life?), an additional question about impacts on job and career since the decision not to be vaccinated and the addition of “Is there anything more you would like to add?” at the end of the interview.”

o The paragraph before the last one on the method sections needs revision. The verbs term needs to be revised. See if you can give more clarity on this paragraph.

RESPONSE: Thank you for this comment – we have made all verbs past tense.

o You mentioned a diagram that ‘summarizes the conceptual relationships between the codes.’ Consider sharing that diagram since that can add clarity or transparency to this section.

RESPONSE: This diagram is Figure 1 and we have now labeled it as such in the text.

Results:

o It is unclear if you collected demographic information about your participant, but if you did, consider adding a table with that information since it will provide valuable information to your study. Also, is information that can contribute to the analysis.

RESPONSE: We did not collect demographic information because of concerns that we might struggle to build trust with this particular group.

o It will be good if you add at the beginning of the result section that you used the ecological model to organize your result section.

RESPONSE: WE have a line that says: “In Figure 1, we present a conceptual diagram which displays our themes and codes using a modified socio-ecological model.” If this is not sufficient, we can add additional information.

o In addition to the participant id, it will contribute to your paper and help to better understand the phenomenon if you add any demographic not identifiable information to the quotes you presented.

RESPONSE: We did not collect demographic information because of concerns that we might struggle to build trust with this particular group. In retrospect, we think they might have been open to disclosing demographics and we agree that this would be helpful in the analysis.

Discussion and conclusion:

o An aspect for consideration is to expand on the discussion or conclusion about the aim of public health. Public health responses aim to benefit groups of people, entire populations, and more during a pandemic. You are evaluating the impact of public health measures on individuals, which is very important. However, it would also be valuable to add the aim of public health efforts, not avoiding the impact of those measures on the individual.

RESPONSE: If we are interpreting this comment correctly, we have added another sentence about novel public health approaches that may be relevant: “Preparedness may also include the use of novel methods such as attitudinal inoculation which is a preventative approach to combating misinformation through pre-exposure to misinformation and delivery of information that allow individuals to refute it (32).”

Best wishes!

---

## [Decision Letter · Decision Letter 1]

9 Aug 2023

“If it wasn’t forced upon me, I would have given it a second thought”: Understanding COVID-19 vaccine hesitancy in an outlier county in the Bay Area, California

PONE-D-23-13758R1

Dear Dr. Brody,

We’re pleased to inform you that your manuscript has been judged scientifically suitable for publication and will be formally accepted for publication once it meets all outstanding technical requirements.

Kind regards,

Mohammad Nusair, Ph.D

Academic Editor

PLOS ONE

Additional Editor Comments (optional):

Reviewers' comments:

Reviewer's Responses to Questions

**Comments to the Author**

1. If the authors have adequately addressed your comments raised in a previous round of review and you feel that this manuscript is now acceptable for publication, you may indicate that here to bypass the “Comments to the Author” section, enter your conflict of interest statement in the “Confidential to Editor” section, and submit your "Accept" recommendation.

Reviewer #1: All comments have been addressed

Reviewer #2: All comments have been addressed

2. Is the manuscript technically sound, and do the data support the conclusions?

Reviewer #1: Yes

Reviewer #2: Yes

3. Has the statistical analysis been performed appropriately and rigorously? 

Reviewer #1: N/A

Reviewer #2: N/A

4. Have the authors made all data underlying the findings in their manuscript fully available?

Reviewer #1: Yes

Reviewer #2: Yes

5. Is the manuscript presented in an intelligible fashion and written in standard English?

Reviewer #1: Yes

Reviewer #2: Yes

6. Review Comments to the Author

Reviewer #1: Important work. All the recommendations were met.

These improve the manuscript. The methods section was improved. Including demographic characteristics of Solano was a plus.

Reviewer #2: Thank you for addressing the recommendations and for the detailed explanations. My only suggestion will be to move the sentences you added to the method section explaining what you had changed after the pilot, " The interview guide was amended..." to the other page when you described how the interview guide was developed and piloted.

Best wishes!

7. PLOS authors have the option to publish the peer review history of their article (what does this mean?). If published, this will include your full peer review and any attached files.

Reviewer #1: No

Reviewer #2: No

---

## [Editor Report · Acceptance letter]

24 Aug 2023

PONE-D-23-13758R1 

“If it wasn’t forced upon me, I would have given it a second thought”: Understanding COVID-19 vaccine hesitancy in an outlier county in the Bay Area, California 

Dear Dr. Brody:

I'm pleased to inform you that your manuscript has been deemed suitable for publication in PLOS ONE. Congratulations! Your manuscript is now with our production department. 

Kind regards, 

on behalf of

Dr. Mohammad Nusair 

Academic Editor

PLOS ONE